# Alcoholic Fermentation as a Source of Congeners in Fruit Spirits

**DOI:** 10.3390/foods12101951

**Published:** 2023-05-11

**Authors:** Damir Stanzer, Karla Hanousek Čiča, Milenko Blesić, Mirela Smajić Murtić, Jasna Mrvčić, Nermina Spaho

**Affiliations:** 1Faculty of Food Technology and Biotechnology, University of Zagreb, 10000 Zagreb, Croatia; damir.stazer@pbf.unizg.hr (D.S.); khanousekcica@pbf.unizg.hr (K.H.Č.); 2Faculty of Agriculture and Food Sciences, University of Sarajevo, 71000 Sarajevo, Bosnia and Herzegovina; m.blesic@ppf.unsa.ba (M.B.); m.murtic-smajic@ppf.unsa.ba (M.S.M.); n.spaho@ppf.unsa.ba (N.S.)

**Keywords:** spirit, yeast, fermentation, aroma compounds, volatiles, threshold, *Saccharomyces*, non-*Sacharomyces*

## Abstract

Fermentation is a crucial process in the production of alcoholic beverages such as spirits, which produces a number of volatile compounds due to the metabolic activities of yeast. These volatile compounds, together with the volatile components of the raw materials and the volatile compounds produced during the distillation and aging process, play a crucial role in determining the final flavor and aroma of spirits. In this manuscript, we provide a comprehensive overview of yeast fermentation and the volatile compounds produced during alcoholic fermentation. We will establish a link between the microbiome and volatile compounds during alcoholic fermentation and describe the various factors that influence volatile compound production, including yeast strain, temperature, pH, and nutrient availability. We will also discuss the effects of these volatile compounds on the sensory properties of spirits and describe the major aroma compounds in these alcoholic beverages.

## 1. Introduction

Fermentation is a metabolic process of conversion of an organic substrate by the action of enzymes. It is a completely natural process that occurs unhindered in nature. It could be said that fermentation is older than the human race because the first fermented products were actually fermented wild fruits. People have been taking advantage of fermentation for thousands of years. Fermentation by selected microbes gives foods new, improved sensory properties, longer shelf life, better nutritional properties, and higher health value [1]. Therefore, people have learned to manage, control, and optimize the fermentation process to achieve better quality of the final product at optimal cost. The simplified definition of alcoholic fermentation (AF) states that it is the conversion of sugar by yeast into ethanol and carbon dioxide with the release of energy. The energy released is necessary for the vital activities of yeast. AF, however, is a very complex biochemical process in which low molecular weight sugars are converted by the action of yeast strains into alcohol and CO_2_ as the main product of alcoholic fermentation, as well as hundreds of other chemical compounds belonging to different chemical classes. Some of them are present in very low concentrations but may have effects on the final product. All of these major and minor chemical compounds of AF form the aroma profile of alcoholic beverages. The aroma profile is the most important characteristic for consumer acceptance and preferences of beverages. It seems that AF plays a crucial role in the production of alcoholic beverages: beer, wine, spirits, and all their derivatives.

Fruit spirits, such as grain and sugar spirits, are a very popular beverage made by fermenting various fruits, grains, and vegetables and distilling the resulting fermented mash. The production of spirits begins with the preparation of the raw material. For this purpose, ripe, high-quality fruits are selected, washed, and cut into small pieces. Fruits provide primary aroma substances that allow to distinguish one fruit spirit from another. It is known that the fruit variety gives a specific sensory note to fruit spirits [2,3,4,5,6,7,8,9,10]. The fruit is then mashed or crushed to release the juice. The production of grain spirits requires saccharification of the starch raw materials to fermentable sugars with enzymes or malt [11]. The next step is fermentation of the raw materials, where yeast consumes the sugars and produces alcohol and a series of volatile compounds. This aroma is called fermentation or secondary aroma. After fermentation, the fermented marc is distilled, and the alcohol and volatile compounds are separated from the fermented solution. Volatile compounds are also formed during the distillation process [12,13]. Some compounds are formed while others are degraded. By separating the fractions, it is also possible to control the amount and type of volatile aromatic compounds in the distillates. Distillation ultimately concentrates the volatile aromatic compounds and enhances their aromatic influence. The aroma produced during distillation is referred to as tertiary aroma. Different distillation processes, such as pot still distillation and column distillation, influence the final flavor and aroma of the spirit [14,15]. During the aging process, additional aroma compounds are formed [16]. The distilled spirit can be aged in wooden barrels or in inert casks, which significantly affects the overall sensory characteristics of the product. The aging process can last from several months to several years, depending on the desired final product. This aging aroma provides crucial opportunities for achieving a positive and distinct sensory characteristic of spirits aged in wooden barrels. The aroma formed during aging is referred to as quaternary aroma. Once the spirit has achieved the desired flavor and aroma profile, it is bottled and sold as a fruit spirit such as *Šljivovica*, William’s pear brandy or apple brandy, grain spirits such as whiskey or vodka, or sugar spirits such as rum or honey brandy.

This chapter compiles information on aroma compounds produced at AF, including the microbiome and conditions for conducting AF. In addition, this article aims to relate the aroma compounds present in spirits to sensory descriptions and their impact on the sensory profile of spirits.

## 2. Spirits Are More than Just Alcohol and Water—They Conclude the Essence of the Scent

Spirit is not consumed for its nutritional value, but primarily for the pleasure it gives. Pleasure is something very personal, and in the case of spirits, it is based mainly on the perception of their sensory qualities. The quality of spirits is evaluated by the consumer on the basis of the sensory perception of their properties. The sensory perception of spirits is the result of the action of a variety of chemical compounds contained in the alcohol–water mixture. Spirits can be sweet and sour in taste, rarely bitter, but they are never salty or umami. This is especially true for unaged distillates without the addition of other flavors or other ingredients.

In aged distillates, the taste of spirits can be more intense in terms of sweetness, bitterness, and acidity due to the action of wood, even in flavored spirits. In contrast to taste, spirits are characterized by their intense odor. The smell is the most important determinant of the sensory quality of spirits and reaches the brain via orthonasal (odor) and retronasal pathways (aroma). Since spirits contain compounds that can evoke trigeminal impressions such as pungent, acrid, and astringent, flavor is the best term to describe the overall sensory perception of spirits. According to ISO 5492, flavor is the complex combination of olfactory, gustatory and trigeminal sensations perceived during tasting.

The sensory impression of fruit spirits depends on hundreds of individual chemical compounds present in an ethanol–water matrix. This very complex mixture contains more than 500 compounds [17] belonging to very different chemical classes: alcohols, esters, aldehydes, ketones, acids, volatile phenols, sulfur compounds, and terpenes. It is essential for the sensory acceptance of the spirit that all the compounds present are in balance with each other and give a pleasant, harmonious impression of beverages. To achieve this, care must be taken at every stage of the spirit production process, from the selection of raw material, through crushing and introduction into fermentation, fermentation conditions, distillation, and finally, the ageing and care of the spirits. Recognizing that the consumer controls consumption, the distilling industry has transformed; producers have become more innovative, introducing new technologies and investing in research. Every effort is made to obtain a product that is safe for health and has the desired sensory characteristics.

## 3. The Research of Spirits Aroma Approach

Numerous studies have been conducted to investigate minor and major flavor compounds in spirits. The number of compounds detected has increased as instrumental methods for flavor investigation have improved, and the composition of spirits is now much clearer [10,17,18,19,20,21,22].

Volatiles are usually analyzed by gas chromatography (GC), which has been greatly improved over time. The GC method has evolved in all areas: column selection, carrier gas selection, temperature programming, injector selection, injector temperature, detector selection, and detector temperature [23]. Aromatic components are now determined using advanced and novel GC methods. Thus, solid-phase microextraction mass spectrometry (SPME-MS) and fast gas chromatography-based electronic nose (GC-E-Nose) have been used and compared in terms of their suitability for discriminating baijiu of different aromatic types and geographical origin [24]. The electronic nose-based on ultrafast gas chromatography was used for the rapid analysis of 24 samples of raw spirits from different cereals [19]. Paolini et al. [22] applied a rapid GC-FID method to identify volatile compounds in spirits. Comprehensive two-dimensional gas chromatography-mass spectrometry analysis (GCxGCxMS) combined with time-off-light mass spectrometry (GC-GC /TOFMS) or with gas chromatography-olfactometry (GC-O) is a powerful technique for the analysis of complex volatiles and allows the identification of minor compounds [25,26,27,28,29,30]. For the differentiation of volatile profiles of eight baijiu distillates, an advanced detection technology involving liquid–liquid microextraction (LLME) combined with gas chromatography-mass spectrometry (GC-MS) and head-space gas chromatography-ion mobility spectrometry (HS-GC-IMS) were used. The profiling of volatiles in soybean Korean brandy was obtained by a novel solid-phase microextraction technique (SPME Arrow technology GC-MS) [31]. Barnes et al. [32] developed a versatile analytical method for the analysis of aroma compounds in various types of distillates produced in micro-distilleries.

As Chambers and Koppel [33] noted, there are many different methods, but all are based on the separation, identification, and quantification of compounds either in headspace or in the actual product matrix.

With the development of instrumental methods, statistical and sensory analyses are also being developed, leading to a better understanding of the relationship between individual chemical compounds and sensory properties of spirits. Hanousek-Čiča et al. [34] used multivariate analysis to classify wine spirits based on their phenolic and volatile compounds. Ivanović et al. [35] performed GC-FID-MS metabolomics to identify 89 compounds in plum spirits and determined the correlation of plum spirit profiles with their sensory properties by dimensional orthogonal projections onto latent structures (OPLS) multivariate analysis. Bajer et al. [36] used OPLS to interpret the correlations between volatile profiles and fruit spirit type. Principal component analysis seems to be the most commonly used statistical analysis to investigate possible groupings of spirit samples based on the aroma compounds studied. Numerous studies with publication dates as recent as possible have been published using PCA as a method to distinguish different spirit samples based on their aroma composition [36,37,38,39,40,41,42,43,44,45,46,47]. Other common methods for clustering different spirits based on their aroma composition include hierarchical cluster analysis or heat map and partial least squares [48,49,50,51,52,53,54]. These techniques provide a clear visual representation of the relationships between aroma compounds and their sensory contributions to spirits or between aroma compounds and the origin and type of spirit.

## 4. Aroma Compounds in Spirts

As mentioned above, spirits contain a large number of aroma compounds, derived from raw materials and formed during the various stages of production. However, AF produces most of the compounds [3] that will be transferred to spirits by distillation.

### 4.1. Aroma Compounds Treated during Alcohol Fermentation 

During the fermentation process, microorganisms convert sugars from raw materials into ethanol, carbon dioxide, and glycerol as the main product of AF and a variety of by-products or intermediates. This complex biochemical process produces hundreds of compounds belonging to different chemical classes, such as alcohols, esters, aldehydes, ketones, acids, and phenols. Although the principle of AF is the same for all alcoholic beverages, the amounts and types of secondary metabolites in beer, wine, and spirits differ significantly, which affects their different sensory properties. Since spirits contain only volatile compounds because they are a product of the distillation process, they are given more attention here. The most abundant compounds in spirits are higher alcohols, followed by esters, acids, and aldehydes (Figure 1). Most of these compounds are fermentative aroma compounds produced by yeast activity. They are mainly the most important flavoring substances in spirits.

#### 4.1.1. Higher Alcohols

With some exceptions in favor of esters [55,56,57], the higher alcohols, also known as fusel alcohols, are the most abundant chemical class in spirits [5,8,58,59,60,61,62]. Yeasts produce higher alcohols during fermentation, either directly from sugars or indirectly from amino acids via the Ehrlich reaction. Amino acids in the wort and fruit must be the primary nitrogen source, and these amino acids (valine, leucine, isoleucine, methionine, and phenylalanine) are taken up by yeast sequentially and slowly during fermentation. After the initial transamination of the amino acids, the resulting α-keto acids are subsequently decarboxylated by the yeast cells and reduced to higher alcohols or acids. A very clear illustration of the Ehrlich pathway was published by [63,64]. Higher alcohols are produced from a sugar substrate via the anabolic pathway. This pathway is involved in the de novo synthesis of branched chain amino acids via their biosynthetic pathway from glucose [65].

The concentrations of higher alcohols in distillates are in the range of 2.5–5.0 g/L of pure alcohol [17]. More than 40 higher alcohols have been detected in spirits and beer [66]. The most abundant alcohols in spirits are the straight-chain alcohols: n-propanol, 3-methyl-1-butanol, 2-methyl-1-propanol, and 2-methyl-1-butanol. Among these alcohols, amyl alcohols are the most represented: 2-methyl-1-butanol (active amyl alcohol) and 3-methyl-1-butanol (isoamyl alcohol) [5]. Hexanol and butanol form the second group of dominant higher alcohols. In Bartlett pear brandies, hexanol was the second most abundant alcohol after 3-methyl-1-butanol [67]; 1-butanol had a very high concentration in melon brandies [68], while the aroma profile of calvados was influenced by a high concentration of 2-butanol [69]. Additionally, 2-phenethyl alcohol is the main aromatic alcohol, and it is responsible for the rose aroma and is recovered at 10%, while the recovery of the above mentioned alcohols is about 90%.

He et al. [57] found 19 higher alcohols in baijiu distillates, and 14 higher alcohols were identified in peach spirits [70], 10 higher alcohols in various berry spirits [71], and 17 in melon spirits [68]. Of nine higher alcohols, none were detected in gin, five were detected in rum, and all nine were present in whisky [32].

Since they are volatile compounds, they are extremely concentrated during distillation. This is why their influence on the flavor profile of spirits is so significant. Even more, the ratio of 2-methyl-1-propanol and 3-methyl-1-butanol can serve as a marker for distinguishing spirits made from different plum varieties [5].

Higher alcohols contribute to a strong, pungent, and fusel aroma profile of spirits when present in large quantities. Rodríguez Madrera et al. [72] reported that the total concentration of higher alcohols > 350 g/hL of absolute alcohol indicates low-quality distillates. In moderate amounts, they contribute to the desirable complexity of spirits. Popovic et al. [58] found that high levels of 1-butanol, 2-butanol, 2-methyl-1-butanol, and 3-methyl-1-butanol resulted in a heavy, unpleasant odor in spirits produced from the “Požegača” variety. A significant contribution to the fusel smell is made by 2-methyl-propanol, 3-methyl-butanol, and 1-butanol, but these are also considered to be the main odor-active compounds in wine distillates and whisky [73]. Tsakiris et al. [17] also noted that in distilled spirits such as brandy, rum, and whisky, fusel alcohols account for most of the common aromatic character. Higher alcohol contents are important for the quality and character of spirits because they serve as precursors for the formation of an important class of desirable compounds, the esters. In particular, n-propanol, n-butanol, and isobutanol are critical to the sweetness and flavor of spirits because they are fundamental components of some esters in spirits, resulting in unique ester flavors [74]. The description and thresholds of alcohols found in spirits are listed in Table 1.

#### 4.1.2. Esters

The chemical class of esters has the most individual compounds in spirits. Of the 140 identified aroma compounds in baijiu distillates, 74 belong to the ester class [57]. Most esters are formed by yeasts as byproducts of sugar metabolism during fermentation. They may also be formed during malolactic fermentation, distillation, and maturation of spirits. Esters in spirits also come from fruits or other raw materials, but due to their low concentration, they have little influence on the overall flavor of the spirit. Esters have a greater influence on the overall profile of spirits than higher alcohols [64].

Esters usually impart a pleasant aroma, contributing to the fruity and floral aroma (Table 2). Although fruit spirits contain almost the same esters, quality spirits differ in their quantity and quality [36]. The most abundant esters are the ethyl esters. Of the 34 quantified esters in raw spirits, 24 belonged to ethyl esters [56]. Chemically, they are classified as either ethyl fatty acid esters or acetate esters. An illustrative metabolic map of the metabolism of ester formation can be found in publication [64].

Ethyl fatty acid esters are formed by the esterification of ethanol with fatty acids during fermentation but also during the distillation process and maturation. The most common are ethyl hexanoate, ethyl octanoate, and ethyl decanoate.

Ethyl esters of short-chain fatty acids such as ethyl butyrate contribute to a pleasant aroma in small amounts but a strong, pungent, cheesy odor in high amounts [79]. In low concentrations, ethyl lactate stabilizes the flavor of the distillate and softens its tart character [80]. Ethyl esters of medium-chain fatty acids (hexanoic acid, octanoic acid, decanoic acid, and dodecanoic acid) are compounds of particular interest in spirits because they impart a fruity and floral aroma and are present in relatively high concentrations [72]. Among them, ethyl hexanoate, which produces a banana-, apple-, or melon-like aroma in spirits, is the most common.

Ethyl esters belonging to the long chain fatty acids such as ethyl dodecanoate, ethyl tetradecanoate, ethyl hexadecanoate, and ethyl octadecanoate contribute to a positive aroma profile when present in low concentrations; in contrast, they are responsible for candle wax and clear tones. The most potent odor-active compounds in raw spirits are ethyl hexanoate, followed by ethyl butyrate, ethyl heptanoate, and ethyl octanoate [56]. In general, the shorter-chain fatty acid esters are more associated with fruity and floral aromas than the longer-chain fatty acid esters [80].

Acetic acid esters are formed during the condensation of acyl-CoA and higher alcohols. The predominant esters in spirits are acetate esters such as ethyl acetate and esters such as 2-methylpropanoate, 2-methylbutyl acetate (fruity flavor), hexyl acetate, ethyl acetate, 3-methylbutyl acetate (banana flavor), and 2-phenylethyl acetate. These esters provide fruity, honey-like, and floral aromas. Among these esters, ethyl acetate predominates in spirits [61,76]. These esters account for more than 80% of the total esters in fruit spirits [14].

A high concentration of esters does not improve the overall flavor quality; on the contrary, a high concentration, especially of ethyl acetate and long-chain fatty acid esters, contributes to an unpleasant solvent tone or fatty and cheesy aroma. They can also mask varietal flavors [81] and make spirits unacceptable.

Over 160 esters have been identified in spirits and wine [17]. Zhang et al. [68] detected 19 esters, which accounted for 31.7% of the total volatile compounds in Malone spirits. Wang et al. [70] identified 44 esters in peach spirits, of which ethyl acetate dominated, followed by phenethyl acetate and 3-methylbutyl acetate (isoamyl acetate). The following ethyl fatty acid esters (in descending order) were most abundant in peach spirits: ethyl 2-methylpropanoate (ethyl isobutyrate), ethyl lactate, ethyl octanoate, and ethyl hexanoate. Ethyl hexanoate, ethyl octanoate, ethyl decanoate, and 2-phenylethyl acetate were the most abundant esters in freshly distilled spirits. However, diethyl succinate and 2-phenylethyl acetate were significant odor-active chemicals in this distillate, and these esters contributed to the sweet rose fragrance of the spirit [73]. Lončarić et al. [82] found in wine spirits that isopropyl tetradecaonate, ethyl hexadecaonate, ethyl cis-9-octadecanoate, and phenethyl acetate were esters responsible for the primary aroma derived from grapes, while ethyl hexanoate and ethyl octanoate were more likely to be developed during the fermentation process. Januszek et al. [83] found nearly 60 esters in apple spirits, with methyl and ethyl esters forming the largest group, to which 26 different compounds were assigned. In Langjiu, a Chinese spirit, 31 ester compounds were found, of which ethyl acetate was the most abundant, followed by ethyl lactate, ethyl propionate, ethyl butyrate, ethyl hexanoate, and ethyl 3-methyl butyrate [76].

#### 4.1.3. Acids

Volatile acids are an important class of flavoring agents in spirits. The most common is acetic acid, which accounts for more than 90 % (v/v) of the total acid content in spirits [14,57]. This acid is produced as a by-product during the AF. The high concentration of this acid in spirits is caused by the contamination of the fermented mash with acetic bacteria and by the late withdrawal of the tail fraction during distillation [84]. Acetic acid is not the only short-chain fatty acid associated with bacterial activity; propionic acid and butyric acid can also be formed by bacterial activity [63]. These acids are characterized by a buttery and cheesy odor and could contribute to the unpleasant taste of spirits (Table 3). In particular, butyric acid may be responsible for the intense sweaty odor when present in high concentrations.

Fatty acids are a natural component of fruits and are also formed during the fermentation process by the activities of bacteria and yeasts. The long-chain fatty acids (C16 and C18) are components of the yeast plasma membrane and are not present in wine as products of yeast, but they are present in spirits due to distillation of the mash [63]. Lambrechts and Pretorius [63] also state that medium-chain (C8, C10, and C12) volatile fatty acids are produced by yeasts as intermediates in the biosynthesis of long-chain fatty acids rather than as a result of acid catabolism. The volatile fatty acids are mainly responsible for the greasy, rancid, and cheesy notes and have a negative effect on the quality of the spirits. The low carbon fatty acids have a greater impact on the overall flavor due to their lower threshold and relatively high concentration in spirits.

If acetic acid is disregarded, the amount of acids in spirits is usually less than 10 mgL ^−1^ of ethanol. Zhang et al. [56] identified 10 volatile acids, of which hexanoic acid was the most important acid in the heart fraction of distillates and accounted for about 50% of the total acids. The highest concentration of hexanoic acids among the nine acids detected in peach brandy was found by Wang et al. [70]. Butanoic acids 2 and 3 methyl are identified as important odorants in Bartlett pear spirits [67], and these two acids were also characteristic of distillates from autochthonous apple varieties [80]. Additionally, 15 acids were detected in baijiu samples, 8 of which were classified as odorants [57].

#### 4.1.4. Carbonyl Compounds

Carbonyl compounds, aldehydes, ketones, and acetyls are also important groups of flavoring agents in spirits. The 2-keto acids, the precursors of aldehydes, are formed as intermediates in both the anabolic and catabolic synthesis of amino acids or higher alcohols. Aldehydes can be secreted, but they can also be reabsorbed and reduced to the corresponding alcohol by yeast during the fermentation process [63].

The predominant aldehyde detected in spirits is acetaldehyde and accounts for 90% of the total carbonyl content in an alcoholic beverage [8]. The content of the other aldehydes is very low. Acetaldehyde is a common fermentation product in yeast fermentations. It is a pyruvate intermediate that serves as a precursor for acetate, acetoin, and ethanol.

When spirits contain a low concentration of acetaldehyde, it contributes to a pleasant fruity aroma, but when its concentration exceeds 125 mg/L [60], it can cause unpleasant rotting odors and pungent and irritating aroma [86]. It is a harmful compound considered to be carcinogenic, so it is important to discard the head fraction during distillation, as acetaldehyde is a typical compound of the first fraction [87]. The 3-methylbutanal (isovaler aldehyde) with a mild apple taste and a sweet and less irritating aroma is more advantageous for the odor of the spirit [73]. The aldehydes 2-butenal and 2-nonenal have been identified as markers for differences between plum spirits from different plum varieties [88].

Normally, diacetyl is formed by yeast metabolism, but it can also be produced at higher concentrations by lactic acid bacteria. At low concentrations, diacetyl has a refreshing effect in spirits, but in general it contributes to the fatty and oily sensory properties [73].

Benzaldehyde is an important aromatic aldehyde, especially for stone fruit spirits. Yeasts are able to utilize benzaldehyde in the presence of glucose, and some yeasts are able to convert it to benzyl alcohol and benzoic acid [63]. The main source of benzaldehyde in stone fruit spirits is the enzymatic degradation of amygdalin. It provides a marzipan-like aroma or a bitter almond aroma.

Ketones are present in very low concentrations. Only four ketones were quantified in crude distillates, with 2-heptanone having the highest concentration [56]. Wei et al. [68] identified only butanone and acetone. Of the ketones in spirits, β-damascenone is a very potent flavor compound due to its very low threshold. It is not considered a fermentation product, although Lloyd et al. [89] demonstrated the formation of damascenone during primary fermentation. In peach spirits, Wang et al. [70] found (E)-β-damascenone to be potentially the most critical aroma compound. During distillation, the concentration of β-damascenone increases [73]. Description and thresholds of carbonyl compounds found in spirits are listed in Table 4.

#### 4.1.5. Volatile Phenols and Other Aroma Active Compounds

Phenolic compounds are much more important for the taste, color, and aroma of wine, while only volatile phenolic compounds are important in unaged spirits. Fruits and grapes usually contain phenolic compounds, and they are particularly important as a source of hydroxycinnamic acid esters, which can be converted to volatile phenols during fermentation. The most abundant are 4-ethylphenol and 4-ethylguaiacol but also 4-ethylcatechol, 4-vinylguaiacol, 4-vinylphenol, and 4-vinilcathecol [64]. These esters are called off-flavor compounds and are responsible for animal, horse sweat, phenol, medicine, and plastic odors and are considered undesirable esters. They are products of *Brettanomyces* infections. In spirits from apples, Spaho et al. [79,80] identified the phenol, 4-ethyl, phenol, phenol, 4-ethyl-2-methoxy (4-ethylguaiacol), and eugenol (clove-like). They had no negative influence on the sensory perception of the apple brandies tested. This means that volatile phenols must be present in odor-perceptible concentrations for them to have a negative effect on the aroma. As already mentioned, fruit brandies also contain numerous primary aroma compounds that originate from the raw material or are released from the wood of the barrel during maturation (Table 5).

### 4.2. Factors Influencing the Development of Aroma Compounds

Many factors affect the chemical composition of the volatile profile of the spirit, including the type and variety of fruit and its quality, AF, distillation, and storage conditions. In the following, we will focus on the fermentation conditions that can influence the development of the aroma. As mentioned above, yeasts are the main producers of ethanol and volatile aroma compounds in the fermented substrate. Therefore, it can be said that yeast is the heart of AF. The selection of appropriate yeast strains is crucial for maximizing alcohol yield and achieving appropriate and attractive sensory characteristics of spirits. Walker [92] suggests and Pauley and Maskell [93] have additionally added to the concept of the ideal distillery yeast, reporting the following characteristics: high ethanol yield; tolerance to ethanol, heat, and high sugar stress; rapid fermentation of available sugars; production of the correct balance of flavor congeners; high viability during storage; and antibacterial and non-flocculating properties. A yeast with these diverse properties does not yet exist, the closest being the *Saccharomyces cerevisiae* species. *Saccharomyces* yeasts currently dominate the fermented beverage industry, but consumer demand for alternative products with different sensory profiles and actual or perceived health benefits is driving the diversification and use of non-*Saccharomyces* yeasts [94]. In this section, the influence of AF parameters, e.g., yeast selection, temperature, pH, nitrogen content, and metal ions, on the volatile profile of fruit distillates is presented.

#### 4.2.1. *Saccharomyces* and Non-*Saccharomyces* Yeasts in Spirit Production

*Saccharomyces cerevisiae* is the yeast most commonly used by humans in many industrial processes such as the production of baker’s yeast, beer, or wine [81]. Of the 210 yeast colonies recovered in Spain’s 6 big distilleries, 144 are *Saccharomyces* strains and 66 are non-*Saccharomyces* strains [95]. 

*S. cerevisiae* is differentiated by its strong fermentative metabolism, good resistance to ethanol produced, high fermentation rate, and high ethanol yield, as well as its capacity to enhance secondary aroma compounds [96]. Selected yeast of *Saccharomyces* contributes to low acetic acid and acetaldehyde production in comparison to indigenous yeasts in spontaneous fermentation. A study by Bovo et al. [97] showed a beneficial effect of the inoculated strain on spirit quality due to the lower content of flavor compounds such as ethyl acetate (glue-like odor) and ethyl lactate (buttery note). Although *S. cerevisiae* produced a lower concentration of total higher alcohols when compared to other non-*Saccharomyces* species [81], there were differences in the concentration of occurred individual higher alcohol. Higher concentrations of isoamyl alcohol and 2-phenylethanol and lower concentrations of C6–alcohols were detected in fermented wine inoculated with *S. cerevisiae* in comparison to non-*Saccharomyces* species [96]. Opposite results are published by Nurgel et al. [98] where selected *S. cerevisiae* decreased the concentration of these alcohols during the fermentation of wine. Additionally, *Saccharomyces* fermentations show differences in the concentrations of individual higher alcohol, most notably 1-propanol [63]. It means that the production of higher alcohols is a strain-specific trait. The same results are proven by considering acetaldehyde, methanol, and ethanol formation in Korean rice spirits where the different *Saccharomyces* strains produced different amounts of mentioned products of alcohol fermentation [99]. 

It is interesting to note that some alcohols—1-hexanol, cis-3-hexen-1-ol, trans-3-hexen-1-ol, and benzyl alcohol—had the same concentration in must before and after fermentation [100]. As a result, those alcohols can be classified as varietal compounds.

Considering the metabolism of volatile acids, *S. cerevisiae* is capable of synthesizing mainly hexanoic and octanoic acids in high amounts but also pentanoic, decanoic, and 3-methylbutanoic acids [81]. The acid concentration is important because it contributes to the formation of desired esters in an alcoholic medium. *S. cerevisiae* produces higher amounts of 2-phenylethyl acetate and ethyl octanoate [98], while the concentration of ethyl acetate is lower in comparison with some indigenous yeasts [86]. 

Due to its good fermentative abilities (above mentioned) and achieving more uniformity of beverages, the starter of selected *S. cerevisiae* strains was favorable in AF. Nonetheless, some non-*Saccharomyces* species have been discovered in recent decades as producers of the better aromatic composition of fermented substrates. The use of non-*Saccharomyces* yeasts in beer and wine is very well described [101,102], in contrast to their use in spirits production. Many studies have shown that non-*Saccharomyces* yeasts can improve the aroma profile of beverages, providing a distinct aroma profile [95,96,103,104,105]. 

Generally, non-*Saccharomyces* species produce a higher concentration of undesirable compounds in the distillate [86]. The importance of non-*Saccharomyces* yeasts stems from the fact that they contain a variety of exogenous enzymes that may influence the formation of highly valued primary aroma compounds from fruit and grapes such as terpenes and norisoprenoids. So, Bovo et al. [97] chose *Saccharomycodes ludwigii* strains aimed to enhance varietal aroma in grape marc spirits, because this yeast has high glycosidase activity. *Torulaspora delbrueckii*, *Metschnikowia pulcherrima*, *Pichia kluyveri*, *Lachancea thermotolerans*, and *Schizosaccharomyces pombe* were used alongside a *S. cerevisiae* as control in the fermentation of grape mash, and the results of volatile profiles of non-*Saccharomyces* yeasts differed significantly from the *S. cerevisiae* control [96]. In production of spirits from dry apple pomace, differences in esters between the yeasts *S. cerevisiae* and *Hanseniaspora uvarum* were detected. The fatty acid ethyl esters (hexanoic, octanoic, decanoic, and dodecanoic acids) were more abundant in the spirits obtained from the *Saccharomyces* strains, in contrast to *Hanseniaspora uvarum* that provided spirits with a higher content in acetic acid esters [72]. Varela [103] mentioned limited research on the role of non-*Saccharomyces* yeasts in the production of tequila, mezcal, and cachaça. It was found that *K. marxianus* provided a wider range of volatile compounds in tequila production compared to *S. cerevisiae*. In addition, *K. marxianus* and *P. kluyveri* produced higher concentrations of ethyl acetate and isoamyl acetate than *S. cerevisiae* during agave fermentation. Sensory analysis showed that mezcal produced with *T. delbrueckii/S. cerevisiae* mixed cultures and with *K. marxianus* cultures was preferred to mezcal produced with pure *S. cerevisiae* cultures, demonstrating a positive influence of non-*Saccharomyces* yeasts on the final sensory profile of agave distillates [106]. Ellis et al. [94] highlighted several key species currently used in the industry, including *Brettanomyces*, *Torulaspora*, *Lachancea*, and *Saccharomycodes*, and emphasized the future potential for using non-*Saccharomyces* yeasts in the production of various fermented beverages. In contrast, Muñoz-Redondo et al. [107] found a decrease in esters and a decrease in fruity descriptors, mainly due to lower concentrations of ethyl esters of medium-chain fatty acids and isoamyl acetate in rosé wines fermented with the commercial non-*Saccharomyces* yeasts *Torulaspora delbrueckii* and *Metschnikowia pulcherrima* in combination with *S. cerevisiae*. However, an increase in other minor esters such as cinnamic acid esters and ethyl esters of branched acids was observed. 

A single microorganism never performs fermentation. Fermentation involves yeasts and bacteria, and the relationship and interaction of these microorganisms are important for aromatic composition.

Xu et al. [108] published that *Lactobacillus*, *Saccharomyces*, *Clostridium*, *Cloacibacterium*, *Chaenothecopsis*, *Anaerosporobacter*, and *Sporolactobacillus* showed strong positive correlations with the main flavor ethyl esters of baijiu. A good example of positive interactions between *Saccharomyces* and non-*Saccharomyces* strains was established between *Pichia anomala* and *S. cerevisiae* as ethyl acetate concentration decreased and isoamyl acetate concentration increased in wine. Another synergistic effect was observed in mixed fermentation using *L. thermotolerans* and *S. cerevisiae*, where glycerol and 2-phenyl ethanol production increased when compared to pure cultures [109]. Similar results were published by Fejzullahu et al. [105] in fruit spirits samples. Samples fermented with a mixture of *Lachancea thermotolerans* and *S. cerevisiae* strains were characterized by a higher concentration of 1-hexanol and 2-phenethyl acetate and 2-methyl-1 propanole. Mixed cultures of *S. cerevisiae* strains with *Torulaspora delbrueckii* increased isoamyl acetate, whereas fermentation of *S.cerevisiae* with *Lachancea thermotolerans* resulted in a decrease in this ester [105]. 

Additionally, the importance of yeast selection and the importance of metabolic products obtained during AF depends on the spirit itself. In the production of refined ethanol and, for example, vodka or light rum, the metabolites are not of critical importance, since in both cases the final product is neutral (pure) ethanol. In these cases, the selected yeast species should be considered from the perspective of high ethanol yield and high tolerance to ethanol and other stresses associated with the fermentation parameters [93]. 

#### 4.2.2. Nitrogen Content

The nitrogen content of raw material affects yeast growth, fermentation rate, and duration and therefore influences generated aroma compounds in fermentation. Fruit and grape include amino acids such as asparagine, glutamine, aspartic acid, glutamic acid, and serine, which can be easily assimilated by yeast, giving relevant higher alcohols and their esters [83]. The apple distillate from the variety with the highest nitrogen content had the most diverse profile of volatile compounds [83]. The addition of nitrogen in the form of ammonium salts is common in some fermentations, such as wine-making. A detailed report by Bell and Henschke [110] discusses the effects of nitrogen addition on grape berry and wine composition and the sensory properties of wine. However, there is a lack of studies examining the effects of nitrogen on the volatile components of fruit distillates. According to the literature, the final concentration of volatile compounds such as acetates of higher alcohols and ethyl esters increases with the increase in initial nitrogen content [111]. In addition, the use of different nitrogen sources, ammonium, or amino acids could affect the aroma profile of wines. The use of amino acids instead of ammonium as the nitrogen source resulted in the production of higher amounts of higher alcohols, indicating their direct catabolic formation via the Ehrlich pathway [112] and higher amounts of acetate esters, 2-phenylethyl acetate and ethyl acetate, and ethyl esters. However, the use of ammonium salt as the sole nitrogen source led to an increase in the production of isoamyl acetate, linalool, 1-octanol, butyric acid, diethyl succinate, and the medium-chain fatty acids hexanoic and octanoic acid [112]. In their study, Barbosa et al. [113] showed that strains of *S. cerevisiae* with nitrogen addition at the stationary phase significantly decreased ethanol and acetic acid formation and significantly increased the following compounds: 2-phenylethanol, ethyl isobutyrate, 2-phenylethyl acetate, ethyl 2-methylbutyrate, and ethyl propionate.

#### 4.2.3. Metal Ions Content

In addition to sugar and nitrogen, yeasts also require an adequate supply of inorganic ions for fermentation. Due to their complexity, yeast cells require a greater number of inorganic elements: macroelements required in millimolar concentrations (phosphorus, sulfur, potassium, and magnesium) and trace elements—microelements required in micromolar concentrations (sodium, calcium, iron, cobalt, zinc, molybdenum, copper, manganese, nickel, and selenium) [114]. There are two basic functions of metal ions that make them essential to the organism: enzymatic (as cofactors for important enzymes) and structural (effects on the stability of important molecules and cell membrane dynamics). Cells obtain the required inorganic ions from the fermentation medium, which usually contains sufficient amounts of inorganic ions for yeast growth, but occasionally, supplementation may be required [115]. The complex composition of substrates, interactions of metal ions with organic compounds, mutual interactions of metal ions, and, in some cases, the use of common transport systems for multiple microelements affect the bioavailability of metal ions and the rate of their accumulation in yeast cells. Although essential, metal ions at higher concentrations can have a toxic effect on yeast cells [116,117]. For this reason, the intracellular concentration of microelements must be optimal and is strictly regulated.

In brewing and distillation fermentation, magnesium and zinc are the most important metal ions [114]. Magnesium is necessary for the growth, metabolism, and activation of various glycolytic enzymes. Zinc acts as an activator of the Zn metalloenzyme ethanol dehydrogenase, and it can stimulate the uptake of maltose and maltotriose in yeast cells; zinc is part of Zn finger DNA-binding proteins involved in the response to general stress. In contrast to the study of the homeostasis of metal ions in yeast cells and the influence on ethanol production, there is little work describing the influence of metal ions on volatile production.

Nicola et al. [118] reported significant changes in volatiles during the fermentation of zinc-enriched beer wort. Yeast cells enriched with zinc produced distillates with increasing concentration of some higher alcohols [118], as well as ethyl caproate and isoamyl acetate [115]. Ribeiro-Filho et al. determined the minimum amount of inorganic phosphate, potassium, and magnesium to support yeast growth and ethanol/aroma formation [119]. Inorganic phosphate, potassium, and magnesium play an important role in the formation of esters and higher alcohols [119,120].

#### 4.2.4. pH Value

Although most of the yeast strains can grow at pH values between 2.5 and 8.5, they are acidophilic organisms, and the optimal pH range can vary from pH 4.0 to 6.0, depending on temperature, the presence of oxygen, and the strain of yeast [121]. pH value influences the yeast growth and fermentation performance by altering cell permeability and cell wall structure. The pH of the substrate also influences the aroma composition of spirits. Acidification treatment of grape pomace showed an influence on changes in yeast–bacteria population ratio and the aromatic profile of the distillate as well [122]. Distillates obtained from acidified grape marc were distinguished by a lower concentration of compounds with potent off-flavor notes. García-Llobodanin et al. [123] published that pear spirits obtained from the fermentation of fruit juice with acidified pH have a higher content of fruity esters and higher alcohols, inter alia, 2-methyl-1-butanol, 3-methyl-1-butanol, and 2-methyl-1-propanol and a lower content of ethyl acetate, as compared with pear spirits made from the fermentation of fruit juice with a native pH. In the production of sour cherry brandy, a decrease in the production yield of volatile compounds was observed when the pH was increased from 3.25 to 3.75 [61]. The fermentation pH was found to be critical for the production of ethyl acetate and ethyl lactate and, thus, for the sensory properties of melon distillates [124]. Plum fruits do not contain major amounts of acids, and their pH is above 3.5 in most cases. Ivanović et al. [35] showed that pH significantly affects the composition of plum brandy. Benzene derivatives were the dominant group of volatile compounds in brandies obtained at pH 3.5. Lowering the pH to pH = 3.0 resulted in an increased content of acetals, monoterpenes, and esters.

#### 4.2.5. Temperature

Yeasts can grow in a wide range of temperatures, but their optimal growth temperature is between 20 and 30 °C. Considering this fact, temperature in AF has a significant impact on yeast growth and metabolism [125,126], including biosynthetic pathways for aroma-enhancing compounds. In the available scientific literature, there are no studies that address the influence of fermentation temperature on the profile of aroma compounds in fruit distillates. However, some parallels can be drawn with the well-studied influence of fermentation temperature on the profile of aroma compounds in wines and beers, as the same process of AF is involved. In addition to fruit variety and yeast strain, AF temperature is an important factor affecting the qualitative and quantitative profile of volatile compounds in fruit spirits, as has also been reported for wines [125,126,127,128,129]. Temperature can alter the rate of AF by affecting the assimilation of nitrogen and sugars and, consequently, has a significant impact on the final quality of the alcoholic beverage [130]. In modern winemaking practice, white wines are fermented at temperatures between 12 and 20 °C, at which higher concentrations of esters responsible for floral and fruity aromas are synthesized [128,131,132,133]. At lower fermentation temperatures, the evaporative loss of volatile compounds is lower, and their stability is higher, resulting in the preservation of desirable primary aromas (flavors of the fruit variety). In addition, low temperature increases the degree of unsaturation of yeast membrane fatty acids and makes the yeast more ethanol tolerant [134]. On the other hand, low temperatures have negative effects on yeast growth and, thus, fermentation performance [90]. Peng et al. [129] compared the concentrations of the main volatile compounds (esters and higher alcohols) in ciders produced at different fermentation temperatures (17, 20, 23, and 26 °C). Based on the results, the fermentation temperature of 20 °C was considered the most suitable for the production of cider. Due to the interactions between yeast strain and temperature, an increase in temperature in the range of 15–25 °C has a positive effect on the formation of volatile compounds and ethanol yield in both wines and distillates [61,135,136] 

The formation of higher alcohols is temperature dependent: while higher temperatures promote the formation of higher alcohols, lower temperatures suppress them. In general, fermentation conditions that promote yeast growth, such as higher temperature, lead to increased production of higher alcohols and their availability, which is required for ester formation, e.g., higher alcohols acetates with pleasant fruity character [137,138] 

According to Pielech-Przybylska et al. [139], fermentation at 18 °C produced fruit spirits with higher concentrations of aldehydes, such as hexanal and benzaldehyde, as well as esters, such as ethyl acetate, isoamyl acetate, and hexyl acetate, than fermentation at 30 °C. *S. bayanus* yeast was more sensitive at a higher temperature, compared to native yeast present on plum in spontaneous fermentation, giving the sharp, acrid taste and unharmonized aroma.

## 5. Aroma Active Compounds

As mentioned earlier, numerous studies have been conducted to investigate the composition of different types of spirits and to measure various influences on the amount of compounds present. However, there is surprisingly little work that determines the content of aroma-active compounds. The mere presence of an aromatic component does not mean that it has a sensory impact on the flavor profile of the spirit. The compound must be present in sufficient quantity to be perceived by humans. Sometimes, the amount must be high for the compound to be perceived; sometimes, the compound can be present in very low concentrations and the consumer can still perceive it. The concentrations of the compound at which it is perceived depend on its odor threshold. This is the lowest concentration of the compound that is detected by smelling (orthonasal) or testing (retronasal). The sensitivity of aroma compounds can be compared based on the threshold concentration [140].

Spirits are a highly complex mixture with hundreds of aroma compounds that provide a fingerprint for the raw material, origin, and method of production. The contributions of these compounds to the aroma profile are not equal. Some of them are more potent and contribute more to the aroma intensity, quality, or, sometimes, aroma character of spirits. The majority of potent aroma compounds have a low odor threshold. To quantify the contribution of each aroma substance to the aroma profile, the odor activity value (OAV) was introduced. The OAVs of the aroma compounds were calculated from the ratio of the concentrations detected in spirits compared to their threshold values, which are usually reported in the relevant literature [57]. This ratio is also referred to as odor value, odor unit, flavor unit, or aroma value.
OAV = C/T
(1)

where C is the concentration of the compound, and OT is its detection odor threshold.

Compounds with high aroma values (greater than 1) may contribute to the aroma profile and actively participate in the overall aroma profile or aromagram of the product. He et al. [57] detected a total of 140 volatile compounds in baijiu samples during distillation by GC-MS, of which 50 contributed to the overall aroma profile. Among the aroma-active compounds, ethyl hexanoate had the highest OAV (7653–46,521), followed by ethyl butanoate, ethyl pentanoate, and ethyl octanoate. In the heart fractions of wine distillates, 45 odor-active compounds were identified by aroma extract dilution analysis (AEDA), but 22 of them had OAVs ≥ 1 [75]. Hong et al. [85] also investigated the contribution of individual aroma compounds to gain deeper insight into their involvement in the aroma diagram of Nongxiangxing baijiu samples. They found 18 aroma compounds with OAVs > 1, and they had strong correlation with flavor attributes and distinct aroma characteristics. The study by Spaho et al. [141] focused on the identification of some aroma active compounds in apple spirits. Among six aroma compounds, they found that ethyl benzoate and benzaldehyde had the largest OAVs. Among the 128 different types of aroma compounds found in litchi (Heiye) wine and distilled spirits, 22 aroma compounds had an OAV ≥ 1 [78]. Considering the OAVs, the authors evaluated the characteristic aromas for lychee distillate based on the elevated concentrations and OAVs of β-damascenone, linalool, ethyl butyrate, ethyl isovalerate, ethyl caproate, trans-rose oxide, and cis-rose oxide. Of the 62 aroma-impacting compounds identified in distilled peach spirit, 14 were considered important for aroma with OAVs ≥ 1 [64]. In traditional Japanese spirits, 14 compounds affect quality, of which β-damascenone has the highest OAV [77].

Although OAV is a good indicator of aroma constituents or key aroma components, it is not a definitive indicator of the actual importance of aroma components in products. As noted by Ferreira et al. [142], there are limitations to OAVs, such as that thresholds are relatively uncertain, OAVs are not strictly related to aroma intensity, and OAVs do not account for the presence of other aromatic compounds that may be enhanced or altered by synergistic or masking effects.

## 6. Conclusions

The profile of volatile compounds in spirits depends on many factors, including the type and quality of raw materials, fermentation, distillation, and, finally, the parameters of the aging process. To produce high quality spirits, fermentation parameters such as pH, temperature, total soluble solids, nitrogen content, and, most importantly, yeast strain must be optimized and the effects of each variable, and their interaction on fermentation and volatile compound formation must be evaluated. The aromatic complexity shown in Table 1, Table 2, Table 3, Table 4 and Table 5 is primarily related to the microbiome, and the composition of the microbiome has a significant impact on the volatiles as well as the final quality of the spirit. Selected yeast strains to produce high quality distillates are on the market today. For many years, yeasts other than *Saccharomyces* were blamed for many of the negative characteristics of fermented pomace, so attempts were made to prevent their action. However, advances in the understanding of AF have led to new findings that non-*Saccharomyces* yeasts can contribute significantly to the increase in compounds that contribute to desirable aromatic properties under controlled conditions. It can be concluded that multi-stage fermentation with selected cultures of *S. cerevisiae* and non-*Saccharomyces* yeast strains should be the most acceptable option with respect to the desired flavor profile of spirits. The literature data support the hypothesis that non-*Saccharomyces* yeasts present during spontaneous fermentation and exhibiting higher activity of various exogenous enzymes enhance the aroma of alcoholic beverages by releasing volatile compounds from some glycosidic forms and forming highly valued primary aroma compounds, which is particularly desirable in fruit distillates. Of course, it should be noted that *Saccharomyces* yeasts are an indispensable component of AF, so the study of non-*Saccharomyces* yeasts must be performed in the context of interactions with *Saccharomyces* species.

## Figures and Tables

**Figure 1 foods-12-01951-f001:**
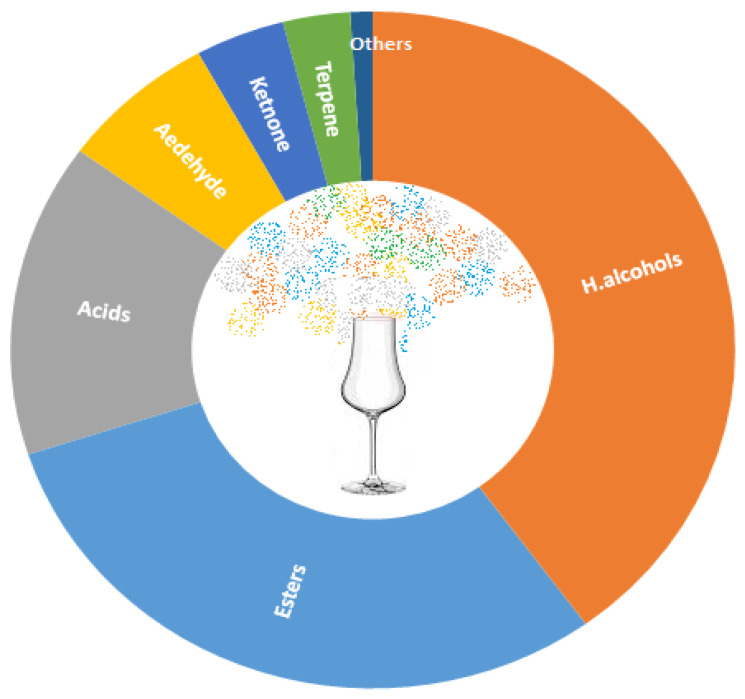
Representation of chemical classes in spirits.

**Table 1 foods-12-01951-t001:** Description and threshold of alcohols found in spirits.

Alcohols	Description	Threshold µg/L	Comment
**Methanol**	Narcotic ether smell, irritation, burning	100,000 [74]	It is not by-product of alcohol fermentation, harmful
**1-Propanol**	Fusel, solvent	54,000 [75]	Formed from threonine
**1-Butanol**	Alcoholic, fruity	2733 [70]	
**2-Butanol**	Alcoholic, pleasant odor	50,000 [61]	
**2-Methyl-1-propanol**	Burnt, fusel, solvent	40,000 [70]	Mainly formed from valine
**3-Methyl-1-butanol**	Fusel, solvent, pungent	56,100.0 [67]	Mainly formed from leucine
**2-Methyl-1-butanol**	Malty, nail polish-like	45,000.0 [67]	Formed from isoleucine
**1-Pentanol**	Fruity, sour, pungent	4000 [70]	
**4-Methyl-1-pentanol**	Alcoholic, plant, green	1000 [75]	
**3-Methyl-1-pentanol**	Alcoholic, plant, fruit, apple	500 [75]	
**2-Phenylethanol**	Flowery, honey-like	2600 [75]	Formed from phenylalanine
**1-Hexanol**	Grassy, almond-like, green	5370 [70]	Originate from raw materials
**2-Hexenol**	Fruity, green	1510 [61]	
**3-Hexanol**	Green	1257 [70]	
**2-Ethyl-1-hexanol**	Rose, green, fruity	1280 [70]	
**3-(Z)-Hexanol**	Fruity, green	1000 [70]	
**2-Heptanol**	Fruity, green	26,600 [70]	
**3-furan-methanol**	Roasted sesame	2000 [61]	
**1-Octen-3-ol**	Mushroom	6.12 [75]	
**1-octanol**	Fruity	1100 [75]	
**3-Nonen-1-ol, (Z)**	Cucumber rind, green, fatty	10 [68]	Important sources of green and fatty flavors in melon spirits
**(E/Z)-3,6-nonadienol**	Cucumber-like, green, fatty,	1.3 [68]	

**Table 2 foods-12-01951-t002:** Description and threshold of esters found in spirits.

Esters	Description	Threshold µg/L	Comment
**Ethyl acetate**	Pineapple, apple-like odor, with astringent, brief taste, glue-like	7500 [73] 32,600 [76]	In small quantity contributes to pleasant fruity aroma. It is discarded with the first fraction during distillation.
**Ethyl propionate**	Banana	19,000 [76]	Desirable aroma compounds in spirits
**Ethyl isobutyrate**	Fruity, citrus, sweet	57.47 [70]	
**Proyl acetate**	Fruity	4740 [76]	
**Isobutyl acetate**	Banana, fruity, apple,	922 [70]	
**Ethyl butanoate**	Apple, pineapple, ripe fruit	81.5 [70]	
**Ethyl 2-methylbutyrate**	Fruity, pineapple, apple	2.2 [77]	
**Ethyl isovalerate**	Fruity, sweet	6.89 [70]	
**Isoamyl acetate**	Banana-like	93.93 [70]	
**Ethyl (S)2-methylbutanoate**	Fruity	0.2 [67]	
**Ethyl pentanoate**	Apple	26.8 [76]	
**Butyl butyrate**	Banana, pineapple	110 [76]	
**Ethyl hexanoate**	Sweet, fruity, green apple	55 [70]	Beneficial for spirit, most abundant among fatty acid esters
**Hexyl acetate**	Green, fruity	1500 [70]	
**Isoamyl butyrate**	Green apple	20 [76]	
**Propyl hexanoate**	Pineapple	12,783.77 [76]	
**Benzyl acetate**	Floral, sweet	270 [70]	In water
**Diethyl succinate**	Fruity Fusel-like and camphor-like	353,193.25 [70]	Can be a consequence of malolactic fermentation
**Ethyl octanoate**	Fruity, pineapple, pear, flowery	12.87 [70]	
**Isopentyl hexanoate**	Pineapple	1400 [76]	
**2-Phenylethyl acetate**	Honey-like, flowery	108 [67]	Beneficial for spirit
**Ethyl lactate**	Dairy, green fruity	128,000 [70]	
**Isoamyl lactate**	Fruity, nutty	131,703.4 [70]	Associated with tail fraction
**Ethyl nonanoate**	Fruity	3150 [76]	
**Ethyl decanoate**	Flowery, fruity, fatty	1120 [70]	
**Ethyl benzoate**	Flowery	1430 [76]	
**Ethyl 9-decenoate**	Fatty	100 [78]	
**Ethyl dodecanoate**	Leaf, fruity, sweet, creamy	1500 [78]	
**Ethyl (E,Z)-2,4-decadienoate**	Pear-like, metallic	1000 [67]	Key congeners for Williams spirits
**Ethyl (E,E)-2,4-decadienoate**	Pear-like, metallic	1800 [67]	
**Ethyl tetradecanoate**	Ether, sweet, flowery	2000 [78]	May cause turbidity and flocculation of distillate
**Ethyl hexadecanoate**	Waxy, oil	1500 [78]	
**Ethyl (E)-cinnamate**	Cinnamon-like	0.8 [67]	
**Ethyl 3-phenylpropanoate**	Fruity, flowery, wine	125 [76]	

**Table 3 foods-12-01951-t003:** Description and threshold of acids found in spirits.

Acids	Description	Threshold µg/L	Comment
**Acetic acid**	Vinegar, acidic	160,000 [70]	It is mainly discarded with the tail fraction during distillation.
**2-Methylpropaonic acid**	Acidic, cheesy	1580 [70]	In free form, they are mainly undesirable in spirits but are important as precursor of ester formation
**Butanoic acid**	Sweaty, cheesy	964 [70]	
**3-Methylbutaonic acid**	Sweaty, dairy	1050 [70]	
**3-Methyl pentatonic acid**	Cheesy	150 [85]	
**Pentatonic acid**	Dairy	390 [85]	
**Hexanoic acid**	Acidic, cheese, sweaty Barbecue	2520 [70]	
**Octanoic acid**	Vegetable, fatty Sweaty cheese	2700 [70] 500 [78]	
**Nonanoic acid**	Coffee, acidity	3560 [85]	
**Decanoic acid**	Rancid, fatty, sweaty	1000 [78]	
**Dodecanoic acid**	Metal	1500 [78]	
**Benzoic acid**	Slightly pleasant and sweet, sour	>10,000 [74]	
**Lactic acid**	Fatty odor, slightly acidic, astringent, thick	<235,000 [74]	

**Table 4 foods-12-01951-t004:** Description and threshold of carbonyl compounds found in spirits.

Carbonyl Compounds	Description	Threshold µg/L	Comment
**Aldehyde acetal**	Grassy odor, fruity, slightly sweet, astringent, refreshing	50,000 to 100,000 [74]	
**Diethyl acetale**	Sweet, fruity	69 [70]	
**Acetone**	Nail polish remover, solvent odor, weakly fruity, pungent	>200,000 [74]	Produced more by *apiculate* yeasts
**Butanone**	Solvent odor, fruity, pungent, sweet	>80,000 [74]	
**2-Octanone**	Hot milk, peanut, green	250 [90]	
**(E)-β-damascenone**	Fruity, flowery, minty, lemon, balsam, honey-like, cooked apple-like	0.4 [67]	Potent aroma compound due to its very low threshold
**Ionne**	Flowery, violet, rosa, spicy	90 [73]	
**Acetaldehyde**	Fruity, overripe bruised apples, sherry-like, stewed apple, pungent and irritating, astringent	19,200 [67]	It is discarded with the first fraction during distillation. Pleasant in low conc. while in high conc. irritating
**Propanal**	Grassy and pungent smell	2500 [74]	Undesirable in spirits, often below their individual perception threshold
**Butanal**	Green leaf, slightly fruity, slightly astringent and bitter	280 [74]	
**2-Methylpropanal**	Grape	800 [73]	
**2-Methylbutanal**	Grassy/sweet	10.6 [67]	
**3-Methylbutanal**	Grassy/sweet, stuffy smell	2.9 [67]	
**Hexanal**	Green/grassy	158.0 [67]	
**(Z)-3-hexanal**	Green/grassy	45.0 [61]	
**(E)-2-nonanal**	Fatty, green	0.6 [67]	
**Nonanal**	Waxy, aldehydic, citrus, fresh, slightly green lemon peel	1 [91]	In wine
**Phenylacetaldehyde**	Honey-like	111 [67]	
**(E,E)-2,4-nonadienal**	Fatty, green	1.1 [67]	
**(E,E)-2,4-decadienal**	Fatty, deep-fried	2.6 [67]	
**Furfural**	Roasted, sweet, Woody, almond	122 [70]	Formed during distillation
**Benzaldehyde**	Bitter almond	4203.1 [70]	Important for stone fruit spirits
**4-hydroxy-3-methoxybenzaldehyde**	Vanilla-like, sweet	22 [67]	
**(E,E)-2,4-nonadienal**	Fatty, green	1.1 [67]	

**Table 5 foods-12-01951-t005:** Description and threshold of terpenes and lactones found in spirits.

**Terpenes**	**Description**	**Threshold µg/L**	**Comment**
**Linalool**	Flowery, citrus	14 [77]	Odoriferous monoterpenes do not show significant changes in their amount during yeast fermentations and originate mainly from fruit Recognition thresholds in 25% (v/v) ethanol solution
**Farnesol**	Flowery	1600 [77]	
**Geraniol**	Flowery, citrus	72 [77]	
**Nerol**	Flowery, citrus, fruity	1900 [77]	
**α-Terpineol**	Minty, medicinal, turpentine, must	7200 [77]	
**D-Citronellol**	Clove	100 [78]	
**β-Citronellol**	Grass/cucumber	100 [73]	
**trans-Nerolidol**	Floral, tea	400 [70]	In water
**trans-β-Ionone**	Sweet, floral	4.5 [70]	
**Eugenol**	Seet, cloves	21 [70]	
**cis-Geraniol**	Rose	300 [78]	
**Neroloxide**	Rose	6000 [78]	
**cis-Rose oxide**	Rose, flowery	20 [78]	
**trans-Rose oxide**	Rose, flowery	20 [78]	
**4-Terpeniol**	Spicy/soil	300 [73]	
**Lactones**	**Description**	**Threshold µg/L**	**Comment**
**cis-Whiskey lactone**	Fruity, cocoa, oakwood	6 [78]	Aroma originated mainly from wood and from fruit
**trans-Whiskey lactone**		20 [78]	
**γ-Butyrolactone**	Fruity	20 [70]	
**γ-Hexalactone**	Tobacco	359,000 [70]	
**γ-Nonanolactone**	Milky notes	90.66 [67]	
**γ-Decalactone**	Apricot and peach	10.87 [70]	

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
