# Peer review of "Alcoholic Fermentation as a Source of Congeners in Fruit Spirits"

_foods, 2023, doi:10.3390/foods12101951_

Round 1

Reviewer 1 Report

This review focuses mainly on fruit-based distilled spirits, and perhaps the title should reflect this. Very little (new) information is presented on cereal-based spirits such as whisky. In fact the majority of the cited references relate to fruit/wine spirits.  Also, the title should refer to compounds (not compound) - although the term "congeners" may be more appropriate. There is too much text to engage the reader, with only one Figure and one (very large) Table. More illustrative material would have been beneficial. There are some strange statements (e.g. "Spirits are characterised by low taste attributes"! and "20 yeast genera" - there are over 280 yeast genera and "contaminant compounds"?).  Overall, although this is a comprehensively referenced Review, I did not find it particularly inspiring.

English language is generally OK, but requires some moderate editing.

Author Response

We thank the reviewer for all comments.

  1. We agree with the comments on the title and have changed it as follows: Alcoholic fermentation as a source of congeners in fruit spirits (line 1).
  2. We agree with the comment about a very long table. We have split the table into several tables so that each chemical class (higher alcohols, esters...) is presented separately in its own table (Table 1-5).
  3. Regarding "more illustrative materials", we have already referred readers to the references where the illustrative materials are shown (line 188), and we have added more references where the illustrative materials are shown (line 237).
  4. We have rephrased "some strange statements" (line 79, 378, 417).

Reviewer 2 Report

1. Lines 28-29, “Fermentation performed by microbes gives foods new, improved sensory properties, longer shelf life, better nutritional properties, and greater health value”. It is inaccurate. Fermentation can also cause food spoilage, sometimes, produce toxin.

 2. In this manuscript, to analyze the influence of alcoholic fermentation parameters on the volatile profile of spirits, yeast strains, nitrogen content, pH, and temperature were described. It is better to include other parameters such as metal ions, and the possible mechanism of these parameters on the production of volatile profile.

Author Response

We thank the reviewer for all comments.

  1. We agree that our statement "Fermentation by microbes gives foods new, improved sensory properties, longer shelf life, better nutritional properties, and higher health value" is inaccurate, and we have reworded it (line 28).
  2. We have added a chapter on the influence of metal ions on the production of volatile components during alcoholic fermentation (lines 494-523).
  3. Due to the introduction of a new chapter, we have added several references (lines 1070-1082), so we have revised the references in the rest of the text and in the tables accordingly.

Round 2

Reviewer 1 Report

The title now more accurately reflects the content of this Review. The sub-division of the previously very large Table is an improvement, but the individual Tables should be re-located to more appropriate sections in the text (eg. following the paragraphs describing the different congener types).  

Other comments:

- Line 82. Many scientists and industrialists would profoundly disagree with the comment that spirits are characterised by low taste attributes. This needs to be re-thought/re-phrased.

- Line 373-4. In what context is Saccharomyces cerevisiae "most commonly used by humans"?

- Line 507. Should be De Nicola et al [reference number?]

Minor language editing required.

Author Response

We thank the reviewer for all comments. We agree that the manuscript is much better now.
1. We have relocated tables in the text after the paragraphs describing the different congener types.
2. Line 82 - we have removed the controversial sentence.
3. We have reworded the sentence as follows: Saccharomyces cerevisiae is the yeast most commonly used by humans in many industrial processes such as the production of baker's yeast, beer or wine.
4. Line 507 - we have inserted the reference number.

5. We have also edited the English language
